# Nutritional Interventions to Improve Asthma-Related Outcomes through Immunomodulation: A Systematic Review

**DOI:** 10.3390/nu12123839

**Published:** 2020-12-16

**Authors:** Lieve van Brakel, Ronald P. Mensink, Geertjan Wesseling, Jogchum Plat

**Affiliations:** 1Department of Nutrition and Movement Sciences, NUTRIM (School of Translational Research in Metabolism), Maastricht University Medical Center, 6200 MD Maastricht, The Netherlands; lieve.vanbrakel@maastrichtuniversity.nl (L.v.B.); r.mensink@maastrichtuniversity.nl (R.P.M.); 2Department of Respiratory Medicine, Maastricht University Medical Center, 6200 MD Maastricht, The Netherlands; g.wesseling@mumc.nl

**Keywords:** asthma, nutrition, immune system, inflammation, lifestyle

## Abstract

Asthma is a chronic inflammatory disease of the airways, characterized by T-helper (Th) 2 inflammation. Current lifestyle recommendations for asthma patients are to consume a diet high in fruits and vegetables and to maintain a healthy weight. This raises the question of whether other nutritional interventions may also improve asthma-related outcomes and whether these changes occur via immunomodulation. Therefore, we systematically reviewed studies that reported both asthma-related outcomes as well as immunological parameters and searched for relations between these two domains. A systematic search identified 808 studies, of which 28 studies met the inclusion criteria. These studies were divided over six nutritional clusters: herbs, herbal mixtures and extracts (*N* = 6); supplements (*N* = 4); weight loss (*N* = 3); vitamin D3 (*N* = 5); omega-3 long-chain polyunsaturated fatty acids (LCPUFAs) (*N* = 5); and whole-food approaches (*N* = 5). Fifteen studies reported improvements in either asthma-related outcomes or immunological parameters, of which eight studies reported simultaneous improvements in both domains. Two studies reported worsening in either asthma-related outcomes or immunological parameters, of which one study reported a worsening in both domains. Promising interventions used herbs, herbal mixtures or extracts, and omega-3 LCPUFAs, although limited interventions resulted in clinically relevant results. Future studies should focus on further optimizing the beneficial effects of nutritional interventions in asthma patients, e.g., by considering the phenotypes and endotypes of asthma.

## 1. Introduction

Asthma is a chronic inflammatory disease of the respiratory system, which affects over 300 million people worldwide [1]. The inflammation is characterized by infiltration of immune cells into the airways, among others the T-helper (Th) cells. In asthma patients, these cells predominantly secrete the Th2 cytokines interleukin (IL) 4, IL-5 and IL-13 [2]. The release of these cytokines activates a cascade of reactions, including mast cell activation and immunoglobulin E (IgE) production. Ultimately, airway inflammation leads to symptoms such as wheezing, cough and shortness of breath [2]. The Global Initiative for Asthma (GINA) provides treatment steps to determine the type of treatment [3]. Frequently, asthma patients have prescribed a combination of short-acting β2-agonists for short-term relief and inhaled corticosteroids in order to suppress the airway inflammation, thereby preventing exacerbations. Compliance to inhaled corticosteroids is generally low and was previously estimated to be between 22% and 63%, whereas short-acting β2-agonists are often used too frequently [4,5]. Noncompliance to inhaled corticosteroids could lead to a gradual worsening of the airway and even systemic inflammation in asthma patients over time [4,6]. This does not only worsen asthma severity, but long-term continuous low-grade systemic inflammation may also contribute to the simultaneous development of disorders related to low-grade inflammation, such as cardiovascular diseases [7]. Therefore, asthma patients could benefit from acceptable and easily applicable strategies in conjunction with pharmacological treatment to decrease airway inflammation and asthma symptoms.

In recent years, there has been a growing interest in the effects of lifestyle and, more particularly, nutrition in the prevention of noncommunicable diseases [8,9,10]. For example, the importance of nutrition in the prevention of common lifestyle-related diseases such as diabetes and cardiovascular diseases is well known and has been described in various reviews and meta-analyses [10,11,12,13]. The role of nutrition in relation to asthma has also been studied extensively. Over the past decades, numerous nutrients, foods, diets or even dietary patterns have been suggested to lower exacerbation rates, improve lung function and asthma control, or even decrease inflammatory markers [14,15]. According to several reviews and meta-analyses, vitamin D [16], omega-3 long-chain polyunsaturated fatty acids (LCPUFAs) [17], and increased fruit and vegetable intake [18] are promising interventions for asthma patients. In line with these observations, the current GINA guidelines state that the use of non-pharmacological strategies on top of asthma medication could contribute to the improvement of asthma control [3]. These guidelines advise clinicians to recommend their asthma patients to follow a healthy lifestyle. More specifically, they recommend to quit smoking [19,20], lose weight in case of obesity (with [21,22] or without [23] bariatric surgery), and consume a diet high in fruit and vegetables. Furthermore, regular physical activity is recommended to reduce cardiovascular disease risk [24] and improve quality of life [25]. However, as mentioned above, asthma severity may be reduced by many more dietary approaches than just by increasing the intake of fruit and vegetables. Moreover, it is relevant to emphasize that most of these interventions potentially affect the immune system, which could explain the decrease in inflammation of the respiratory system and the consequent reduction in asthma complaints. An example is a suggestion made in a review that vitamin D could have a direct effect on the immune system, since a variety of immune cells involved in asthma pathology express the vitamin D receptor [26]. However, despite promising results of vitamin D interventions [16], data are inconsistent. This may at least partly be related to the different doses of vitamin D supplemented in the various studies [26,27]. Therefore, a better understanding of mechanisms underlying the immunomodulatory effects of nutritional components is needed to support dietary approaches to improve asthma control. However, a systematic evaluation of randomized controlled trials studying these effects is missing. Therefore, the aim of this review is to provide an overview of nutritional interventions in asthma patients and reported asthma-related outcomes as well as immunological parameters and to search for possible relations.

## 2. Materials and Methods

### 2.1. Search Strategy

The preferred reporting items for systematic review and meta-analyses (PRISMA) checklist was used to structure this systematic review [28]. The aim of the search was to find controlled intervention studies in which the effects of a dietary intervention on asthma-related outcomes as well as on immunological parameters in asthma patients were reported. Three databases (Medline, Embase and the Cochrane Controlled Register of Trials) were used to conduct the search strategy. The abstracts of papers present in these databases were searched in August 2019 without any restriction on publication date. The following search string was used: asthma and immune system or immune function or immunology or inflammation or inflammatory or immunity and trial or clinical study or intervention or RCT and nutrition* or food or diet* or supplement* or micronutrient* or antioxidant*.

### 2.2. Selection of Studies

After the removal of duplicates, two researchers (LvB and JP) screened the articles that were retrieved from the databases. The screening consisted of two rounds. First, titles and abstracts were screened to determine potential eligible papers. Second, all these papers were read and included in the systematic review if they met all inclusion criteria. Relevant papers present in the reference lists of selected articles were screened as well. The following inclusion criteria were used: (1) randomized-controlled trials, (2) written in English, (3) scientific papers, (4) publication in a peer-reviewed journal, (5) used a nutritional intervention, (6) the subjects used medication for allergic asthma, (7) at least one immunological parameter was reported, (8) at least one asthma-related outcome was reported. Studies were excluded if they did not have a control group or were not randomized. Additionally, conference abstracts or posters were excluded. Differences in selection by the two researchers were solved by discussion. If two articles referred to the same study, both were included in the review, provided that both articles presented either different asthma-related outcomes or immunological parameters. Data of articles describing the same study were merged and presented as one study in tables. In this systematic review, this occurred twice and has been mentioned as a footnote of the corresponding table.

### 2.3. Data Extraction

After the second screening, data were extracted from the eligible papers and transferred to Excel. The following characteristics were extracted: (1) study information (first author, year of publication, study design, duration, subgroups, intervention, type of asthma, participants’ health status, medication use, dietary requirements during the study) (2) baseline characteristics (number of subjects, gender, age, body mass index (BMI)), (3) immunological parameters measured, and (4) asthma-related outcomes measured. For each paper, data were extracted for the experimental and control groups separately. If outcome parameters were only presented in a figure, corresponding means or medians were estimated using a pixel ruler [29]. If not presented in the paper, within-group changes were calculated by subtracting the mean or median outcome of the baseline measurement from the mean or median of the final measurement in that group for both parallel and crossover studies [30,31,32].

For asthma-related outcomes, a wide variety of parameters was reported. Therefore, we decided that the three most reported questionnaires (asthma control test (ACT), asthma control questionnaire (ACQ) and asthma quality of life questionnaire (AQLQ)) and the three most reported lung function parameters (forced expiratory volume in one second (FEV_1_), forced vital capacity (FVC) and peak expiratory flow (PEF)) were the focus of this review. Moreover, a study was classified to result in improved asthma-related outcomes if at least one of these parameters was significantly improved as compared to control treatment. For immunological outcomes, there was an even wider variation in outcome parameters. Therefore, we decided to focus on markers that were reported by at least five different studies. A study was classified to result in improved immunological outcomes if at least one of these five parameters was significantly improved as compared to control treatment. If a study did not report one of the three most used asthma questionnaires or lung function parameters, results were still included in the systematic review and listed under the heading “other” in the corresponding tables. The same applies to studies that only used immune markers that were not measured in five or more studies. Interventions that observed statistically significant changes in both asthma-related outcomes and immunological parameters were used to search for possible relations between these two outcome domains.

FEV_1_ and FVC were reported as the percentage of the predicted value unless the original paper only reported these outcomes in liters. If FEV_1_ or FVC were reported in milliliters, the unit was transformed into liters. PEF was also reported as the percentage of the predicted value unless the original paper reported PEF in L/min. The total score of the AQLQ was calculated from domain scores if the total score was not presented in the original paper. Units for the immunological parameters IgE, IL-10 and C-reactive protein (CRP) were transformed into IU/mL, pg/mL and mg/L, respectively. Units for eosinophils in blood were transformed into 10^9^ cells/L unless the original paper reported eosinophils as a percentage of total leukocytes.

### 2.4. Methodological Quality Assessment

The methodological quality of the trials was assessed by calculating the Jadad score [33]. Studies received a total score ranging from 0 to 5 based on methodological aspects, including randomization, blinding and description of withdrawals. Outcomes are described in Appendix A.

## 3. Results

The search resulted in 808 records, and six other papers were identified through other sources. In the end, 30 articles were included in this review after removal of duplicates, the title and abstract screening (first screening round) and the full-text screening (second screening round) (Figure 1). These articles described 28 individual studies.

These 28 studies were clustered based on the nutritional interventions used. The following six clusters (with their respective number of articles) were formed: herbs, herbal mixtures and extracts (*N* = 6); supplements (*N* = 4); weight loss (*N* = 3); vitamin D3 (*N* = 5); omega-3 LCPUFAs (*N* = 5); and whole-food approaches (*N* = 5). An overview of the clusters and selected studies with their characteristics is presented in Table 1.

### 3.1. Herbs, Herbal Mixtures and Extracts

*Asthma-related outcomes*: The results for the cluster “Herbs, Herbal Mixtures and Extracts” are shown in Table 2. Lung function parameters were reported in five studies in the “herbs, herbal mixtures or extracts” cluster. Saffron, *Nigella sativa* or an extract *of B. serrata* gum resin and *A. marmelos* fruit caused an increase in these parameters as compared to the control group [36,40,41]. In contrast, no effect of *Nigella sativa* was found on airway function in another study [39]. Only within-group changes were reported in a study using an extract of propolis. Therefore, no statement could be made if the change of the intervention group differed significantly from the change in the control group [38]. Improvements in asthma control as measured via the ACT were reported in three studies using *Nigella sativa* [35,39,40]. Furthermore, an extract of *B. serrata* gum resin and *A. marmelos* fruit improved asthma-related quality of life [41].

*Immunological parameters*: *Nigella sativa* oil did not change fractional exhaled nitric oxide (FeNO) [40] but caused a decrease in the number of eosinophils in blood [39].Th1 cytokines were reported in four studies. *Nigella sativa* and an extract of *B. serrata* gum resin and *A. marmelos* fruit increased interferon-γ (IFN-γ) [34,40,41]. Th1 cytokines were reported in a study using an extract of propolis, but between-group changes were not presented. Therefore, no statement could be made if the change of the intervention group differed significantly from the change in the control group [38]. Th2 cytokines or IgE were reported in four studies. *Nigella sativa* and an extract of *B. serrata* gum resin and *A. marmelos* fruit caused decreases in these parameters [34,41]. In contrast to the *Nigella sativa* intervention by Barlianto et al. (2017) [34], two other *Nigella sativa* interventions did cause a significant effect on Th2 cytokines or IgE [39,40]. Treg cytokines were analyzed in two studies. *Nigella sativa* had no effect on these parameters [40]. Only within-group changes were reported in a study using an extract of propolis [38]. Proinflammatory markers were reported in three studies and only decreased after saffron supplementation [36]. *Nigella sativa* did not change these markers [40]. Only within-group changes were reported in a study using an extract of propolis [38].

*Overlap between both domains:* Five out of six studies from this herb, herbal mixture, or extract cluster showed significant improvements in both asthma-related as well as immunological parameters, compared to the control group [34,35,36,37,39,40,41]. *Nigella sativa* (*N* = 3) increased ACT score in children with 1.9 points [34,35], and in adults with 1.3, 1.4 [40] or 2.1 points [39]. It also increased PEF variability with 8.2 and 5.8 L/min in adults, depending on the dose used [40]. *Nigella sativa* also increased IFN-γ with 7.8 pg/mL in children [34,35] and with 0.5 or 0.9 pg/mL in adults [40]. It decreased IL-4 with 0.3 pg/mL in children [34,35] and eosinophils in blood with 65 cells/µL in adults. Saffron (*N* = 1) increased FEV_1_ with 5% and FVC with 1% and decreased CRP with 37.5 ng/mL in adults [36,37]. A mixture of *B. serrata* gum resin and *A. marmelos* fruit (*N* = 1) improved PEF with 46.2 L/min, the AQLQ score with 0.5 points, increased IFN-γ with 7.6 pg/mL and decreased IL-4 with 0.4 pg/mL in adults [41].

### 3.2. Supplements

*Asthma-related outcomes*: The results for the cluster “Supplements” are shown in Table 3. All six studies that used supplements as intervention reported lung function parameters. Soy isoflavone supplementation decreased FVC [44]. Using vitamin E, tomato juice, or a tomato extract did not cause changes in lung function [43,45]. Another study with vitamin E supplements only reported within-group changes. Therefore, no statement could be made if the change of the intervention group differed significantly from the change in the control group [42]. No effects were found of soy isoflavones on ACT score [44] or a tomato extract or juice on ACQ score [45].

*Immunological parameters*: Two studies reported the effect of their intervention on FeNO, but no effects were reported for a tomato extract or tomato juice [45]. Soy isoflavone supplementation even worsened FeNO concentrations [44]. The number of immune cells in sputum or blood was reported by two studies. Tomato extract and tomato juice decreased the number of immune cells in sputum [45]. Soy isoflavones had no effect [44]. Two studies reported Th2 cytokines or IgE, and one study reported proinflammatory markers. Vitamin E did not change IgE levels [43]. Another study using vitamin E supplements only reported within-group changes. It is unknown if the change of the intervention group differed significantly from the change in the control group [42]. Soy isoflavones did not change proinflammatory markers [44].

*Overlap between both domains*: None of the studies reported improvements in both asthma-related outcomes and immunological parameters. In adults, soy isoflavones worsened both asthma-related outcomes and immunological outcomes, as FVC was decreased by 0.1 L and FeNO increased with 4.9 ppb compared to the control group [44].

### 3.3. Weight Loss

*Asthma-related outcomes*: The results for the cluster “Weight Loss” are shown in Table 4. Lung function parameters were reported in all three studies of the weight loss cluster. Low caloric intake combined with weight loss medication (sibutramine and orlistat) caused a reduction in BMI of 5.3 kg/m^2^ in the intervention group and improved FVC compared to the control group [46]. A high protein and low glycemic index diet, high-intensity interval training, or a combination of this diet and high-intensity interval training caused weight losses of 2.3 kg, 1.0 kg and 3.1 kg, respectively [48]. Combining energy restriction and counseling sessions resulted in a weight loss of 3.4 kg and a reduction in BMI z-score of 0.2 [47]. No changes were found in lung function parameters in these two studies [47,48]. Additionally, asthma control was evaluated in all studies. Low caloric intake combined with weight loss medication and energy restriction together with counseling sessions improved asthma control [46,47]. The study evaluating diet and high-intensity interval training reported an improvement in asthma control in the group that combined the diet with high-intensity interval training [48]. Asthma-related quality of life was increased after this same diet combined with high-intensity interval training [48], but not after energy reduction combined with counseling sessions [47].

*Immunological parameters*: FeNO was reported as an immunological outcome in all studies, but no significant improvements were found. The number of immune cells was also evaluated by all three studies. Energy reduction, combined with counseling sessions, decreased sputum lymphocyte numbers [47]. The other interventions did not change immune cell numbers [46,48]. Low caloric intake, combined with weight loss medication, did not change IgE levels [46]. Proinflammatory markers were reported in all studies. Only energy reduction in combination with counseling sessions decreased CRP [47].

*Overlap between both domains*: Energy reduction in combination with counseling sessions in children significantly decreased ACQ score by 0.6 points, decreased sputum lymphocyte numbers with 0.1 × 10^6^ cells/mL, and decreased CRP with 1.1 mg/L compared to the control group [47].

### 3.4. Vitamin D3

*Asthma-related outcomes*: The results for the cluster “Vitamin D3” are shown in Table 5. Lung function parameters were evaluated in four out of five studies from the vitamin D3 cluster, but no significant improvements were found [50,51,52,53]. The effect of vitamin D3 on asthma control was also evaluated in four studies, but none found significant improvements [50,51,52,53]. The effect of vitamin D3 on asthma-related quality of life was reported in two studies, although no changes were reported [51,52]. Finally, Bar Yoseph et al. (2015) did not find an effect of vitamin D3 on the provocative dose causing a 20% fall in FEV_1_.

*Immunological parameters*: FeNO was reported in three studies, but no changes were found [49,51,53]. Only a single dose of 400.000 IU vitamin D3 decreased sputum eosinophilia after adjustment for baseline values [51]. Other interventions did not change the number of eosinophils in sputum [50] or blood [49]. Th2 cytokines and IgE were reported twice [49,51], as well as Treg cytokines [49,52]. No changes in these parameters were reported. Two studies reported proinflammatory markers. One study using a daily dose of 2.000 IU vitamin D3 reported an increase in CRP [52], whereas no effect on CRP was found in another study using the same dose of vitamin D3 [49].

*Overlap between both domains*: None of the vitamin D3 interventions resulted in improvements in both asthma-related outcomes and immunological parameters.

### 3.5. Omega-3 LCPUFAs

*Asthma-related outcomes*: The results for the cluster “Omega-3 LCPUFAs” are shown in Table 6. Lung function parameters were evaluated in all five studies of the omega-3 LCPUFA cluster [54,55,56,57,58]. PEF increased after using a lipid extract of the New Zealand green-lipped mussel [54,56]. Omega-3 LCPUFA supplementation, an omega-3 LCPUFA rich diet or an omega-3 LCPUFA enriched fat blend did not improve lung function parameters [55,57,58]. ACQ score did not change after omega-3 LCPUFA supplementation [57].

*Immunological parameters*: FeNO was reported by three studies and decreased after using an extract of the New Zealand green-lipped mussel and a fat blend enriched with omega-3 fatty acids [56,58]. Another study did not find any changes [57]. Furthermore, the number of sputum eosinophils was not changed after increasing omega-3 LCPUFA intake [55,58]. Finally, Emelyanov et al. showed that an extract of the New Zealand green-lipped mussel decreased exhaled H_2_O_2_ [54].

*Overlap between both domains*: Two studies that used lipid extracts of the New Zealand green-lipped mussel as intervention found significant improvements in asthma-related outcomes as well as immunological parameters in adults, compared to the control group [54,56]. Combined morning and evening PEF increased with 21.8 L/min and FeNO decreased with 9.9 ppb in one study [56], whereas morning PEF increased with 80.4 L/min and exhaled H_2_O_2_ decreased with 0.1 uM in the other study [54].

### 3.6. Whole Food Approaches

*Asthma-related outcomes*: The results for the cluster “Whole Food Approaches” are shown in Table 7. Lung function parameters were reported in all five studies using a whole-food approach intervention [59,60,61,62,63]. Consuming a high antioxidant diet increased FEV_1_ and FVC [63]. The other interventions did not change lung function parameters. Additionally, asthma control was evaluated in all studies. However, no changes in ACT or ACQ score were found. The Mediterranean diet did not improve asthma-related quality of life [60,61].

*Immunological parameters*: FeNO was reported in four studies [59,60,62,63]. Consuming two meals with fatty fish per week as part of the Mediterranean diet decreased FeNO [60]. A nutrient-dense bar [59], broccoli sprouts [62] and a high antioxidant diet [63] did not have an effect on FeNO concentrations. The Mediterranean diet and a high antioxidant diet did not change Th1 cytokines [61,63]. In the same study on the Mediterranean diet, no changes in Treg cytokines and immune cell count were found [61]. Using a nutrient-dense bar or broccoli sprouts did not change Th2 cytokines or IgE [59,62]. Proinflammatory markers were not changed after using a nutrient-dense bar, the Mediterranean diet, a high antioxidant diet or broccoli sprouts as an intervention [59,61,62,63].

*Overlap between both domains*: None of the studies from this whole-food approach cluster found improvements in asthma-related outcomes and immunological outcomes simultaneously.

### 3.7. Effect Sizes in the Context of Minimal Clinically Important Difference

Eight studies found changes in asthma-related outcomes as well as in immunological parameters, which may indicate a link between immunological parameters with asthma-related outcomes. The magnitude of changes in asthma-related outcomes in comparison to the minimal clinically important difference is shown in Figure 2. In three of these eight studies, the changes as compared to those of the control group had an effect size that exceeded the minimal clinically important difference. These three interventions were part of the following clusters: herbs, herbal mixtures and extracts (*N* = 1), weight loss (*N* = 1) and omega-3 LCPUFAs (*N* = 1) and showed effects on PEF (extract of *B. serrata* gum resin and *A. marmelos* fruit; lipid extract of the New Zealand green-lipped mussel), ACQ (energy reduction combined with counseling) and AQLQ (extract of *B. serrata* gum resin and *A. marmelos* fruit). A minimal clinically important difference of FVC could not be found in the literature.

## 4. Discussion

The aim of this systematic review was to provide an overview of studies that examined the effects of nutritional interventions in asthma patients on both changes in asthma-related outcomes as well as immunological parameters and search for possible relations. The reason for this approach was that we hypothesized that nutritional interventions might affect asthma severity via modulation of the immune system. Therefore, interventions that improved asthma-related outcomes, as well as immunological parameters, were considered as indications of a link between these two outcome domains, without claiming causality. Current guidelines for asthma patients do contain lifestyle recommendations, but only to a limited extent. It is only advised to increase fruit and vegetable intake and to maintain a healthy weight in order to improve asthma-related outcomes. Based on the results of our systematic review, it may be interesting to consider whether these lifestyle recommendations could be extended in the future. As shown in Table 2, Table 3, Table 4, Table 5, Table 6 and Table 7, fifteen out of 28 controlled dietary intervention studies reported an improvement in at least one asthma-related outcome or immunological parameter. However, also two studies reported a worsening in one of the domains, of which one study using soy isoflavones observed a worsening in both domains. With respect to our hypothesis that nutritional interventions likely affect asthma severity via modulation of the immune system, we identified eight studies that showed a simultaneous improvement in asthma-related outcomes and immunological parameters. These studies used *Nigella sativa* (*n* = 3), saffron (*n* = 1), an extract of *B. serrata* gum resin and *A. marmelos* fruit (*n* = 1), energy reduction in combination with counseling sessions (*n* = 1) or a lipid extract of the New Zealand green-lipped mussel (*n* = 2) as an intervention. Regarding the clinical relevance of these results, we showed in Figure 2 that three out of the eight studies were able to find clinically relevant changes in asthma-related outcomes. Clinically relevant changes were found in lung function parameters, asthma control and quality of life.

In the context of identifying nutritional interventions that improve asthma-related outcomes via immunomodulation, the “herbs, herbal mixtures and extracts” cluster showed the most consistent and promising results. When interpreting the effects of this cluster, however, it should be noted that exact concentrations of the active compounds of herbal extracts are not always known, and concentrations of extracts could be variable. The use of *Nigella sativa* resulted in an improvement in asthma control and PEF, which was accompanied by a reduction in eosinophils numbers in blood, an increase of IFN-γ and an increase in IL-4 [34,35,39,40]. An intervention with saffron increased FEV_1_ and FVC and simultaneously decreased CRP levels [36,37]. An increase in PEF and asthma-related quality of life was observed after using an extract of *B. serrata* gum resin and *A. marmelos* fruit. These changes in asthma-related outcomes occurred simultaneously with an increase in the Th1 cytokine IFN-γ and a decrease in the Th2 cytokine IL-4 [41]. These results suggest that the improvement in asthma-related outcomes after using these herbal interventions is mediated by affecting the Th1/Th2 balance, thereby dampening the Th2 driven pathological process. Since the Th1/Th2 balance in asthma is disturbed as asthma is characterized by Th2 mediated inflammation, asthma patients could benefit from interventions that increase Th1 activity, thereby contributing to restoring the Th1/Th2 balance [64]. Based on mouse models, it has been suggested that increased production of Th1 cytokines, such as IFN-γ, could contribute to a decrease in immune cell infiltration in the lungs and eventually to a decrease in local inflammation [64,65,66]. This suggested mechanism is supported by the findings in this review and is in accordance with previous research. For example, the main constituent of *Nigella sativa* oil, thymoquinone, stimulates IFN-γ production and decreases IL-4 production in animal models of asthma [64,67,68]. Moreover, lowering Th2 cytokine concentrations is part of the suggested mechanism underlying the effects of crocin, which is the main active component of saffron [69]. However, there are additional pathways that can be modulated by thymoquinone [70] and crocin [71], which indicates that herbal interventions could reduce asthma-related complaints via several mechanisms.

The second cluster that showed promising results was the omega-3 LCPUFA cluster. Two studies using a lipid extract of the New Zealand green-lipped mussel both showed an increase in PEF, which occurred simultaneously with a decrease in FeNO concentrations in one study [56] and a decrease in exhaled H_2_O_2_ in the other study [54]. FeNO and exhaled H_2_O_2_ are both markers for airway inflammation and can be analyzed in exhaled breath, and correlate positively with eosinophils in induced sputum [72,73]. Furthermore, FeNO can provide information on the asthmatic state that is consistent with other biomarkers for inflammation in asthma [74] and also decreases after treatment with inhaled corticosteroids [75,76]. The results found in this cluster suggest that the effects of interventions with omega-3 LCPUFAs on asthma-related outcomes are mediated by a decrease in airway immune cell infiltration and local inflammation. These anti-inflammatory effects of omega-3 LCPUFAs and their mediators, such as resolvins and protectins, are generally acknowledged and are in line with earlier findings [77]. Several findings in mouse models confirm the results of the studies described in this review. For example, a study in mice suggested that administration of resolvin E1 to an experimental asthma model resulted in an increased IFN-γ production and a decrease in the proinflammatory lipid mediator leukotriene B4 [78]. Leukotriene B4, as well as other leukotrienes, may be involved in immune cell recruitment in lung tissue of asthma patients [79,80]. Also, leukotriene B4 production in neutrophils was previously found to be reduced after human subjects were supplemented with omega-3 LCPUFA and omega-6 short-chain fatty acids [81]. Another study in Fat-1 transgenic mice, which can endogenously produce omega-3 from omega-6 fatty acids, showed that Fat-1 mice had decreased concentrations of Th2 cytokines IL-5 and IL-13 in their lung tissue as compared to wildtype mice [82]. In short, these results suggest that mediators of eicosapentaenoic acid (EPA) and docosahexaenoic acid (DHA) stimulate IFN-γ production by Th1 cells and inhibit the production of Th2 cytokines and other proinflammatory markers such as leukotrienes, at least in asthma conditions. This could ultimately lead to an inhibition of immune cell recruitment into the airways and therefore contribute to an improvement in asthma-related outcomes. However, not all omega-3 LCPUFA intervention studies in this review showed improvements in asthma-related outcomes that were mediated by the immune system. Strikingly, the three studies with the lowest daily doses of EPA and DHA were the studies that showed beneficial effects on asthma-related outcomes as well as immunological parameters. These studies used intakes between 120 and 300 mg/day. Additionally, one study using a daily dose of EPA and DHA of 630 mg only reported a decrease in FeNO concentrations, but not in asthma-related outcomes [58]. Studies using higher daily intakes of 780 mg or even 1200 mg did not report changes in asthma-related outcomes or in immunological parameters [55,57]. The Food and Agriculture Organisation/World Health Organization set an acceptable macronutrient distribution range for omega-3 LCPUFA between 250 and 2.000 mg/day [83]. Based on the results of this systematic review, asthma patients may benefit from the immunomodulatory effects of omega-3 LCPUFAs when doses on the lower side of this range are consumed. Regarding the effects of higher doses, Yin et al. (2009) showed that supplementing the diet of mice with high doses of fish oil (2% *w/w* and 4% *w/w*) increased the levels of Th2 cytokines IL-5 and IL-13 in lung tissue [84]. It is possible that this mechanism explains why the studies using higher doses of EPA and DHA described in this review did not find the improvements in asthma-related outcomes as observed with the lower doses. The underlying mechanism of the dose-response effect of omega-3 LCPUFAs in asthma patients should be explored further.

The least promising clusters were the supplement cluster, the vitamin D3 cluster and whole-food approaches. Since many different interventions were part of the supplement and whole-food approach clusters, we cannot conclude that these types of interventions will never affect asthma-related outcomes. For vitamin D3, the evidence remains contradictory. We found no indications that vitamin D3 is beneficial for asthma patients. However, future studies should take into account the vitamin D status at the start of the study, dose, timing of the dose (single dose versus multiple doses) and parameters that could be affected by vitamin D3, e.g., immune cells that express the vitamin D receptor.

The current GINA guidelines for asthma state that a diet high in fruit and vegetable intake, as well as weight loss, could improve asthma-related outcomes. Indeed, all interventions of the weight loss cluster had a positive effect on asthma-related outcomes and, more specifically, asthma control. This agrees with the systematic review of Okoniewski and colleagues, who reported that weight loss improved a variety of asthma-related outcomes [85]. However, these beneficial effects hardly coincided with improvements in the response of the immune system in our current review. An explanation for this could be that obesity-related asthma has lower eosinophilic inflammation compared to other phenotypes, and therefore the asthma-related changes after weight-loss were mediated by other mechanisms [86]. It has been suggested that adipokines such as leptin and adiponectin could be involved, which can directly affect airway reactivity [87]. Furthermore, two of the whole-food approaches targeted fruit and vegetable intake of asthma patients [62,63]. Only one of these studies, which used a high antioxidant diet, reported changes in asthma-related outcomes. None of these studies, unfortunately, reported changes in immunological parameters. The effect of fruit and vegetable intake on asthma has been described previously and indeed improved asthma-related outcomes, which has been attributed to their high anti-oxidant and fiber contents [18]. The high antioxidant capacity of fruit and vegetables, as well as the short-chain fatty acids formed from fiber by the microbiota have been suggested to reduce airway inflammation [18]. These suggestions are, unfortunately, not in accordance with the results of this review. However, the number of studies included in this review evaluating fruit and vegetable intake was limited, and these studies mainly addressed the short-term intake of fruits and vegetables. It remains unclear if fruit and vegetables may influence asthma-related outcomes via other routes than the immune system.

Finally, as shown in this review, studies using the same nutritional intervention do not always find similar effects on asthma-related outcomes and immunological parameters. This could be explained by several factors that influence the success of nutritional interventions in asthma patients in general. These factors that were unintendedly part of the study populations may certainly have influenced the interpretation of the results described. An example of such a factor is asthma severity at baseline. Scott and co-workers found that a 10-week weight loss intervention was more successful in participants with more severe asthma at baseline. They suggested that asthma severity could have been a motivator for this group of patients since the burden of the severity of asthma was a motivation for weight loss [88]. Moreover, nutritional status at the start of the intervention may influence the success of the nutritional intervention in asthma patients. Poor diet quality, which may lead to deficiencies of several micronutrients, has been associated with severe asthma [89]. Therefore, dietary interventions might be more successful in asthma patients that have deficiencies at baseline [16]. In addition, study duration could have influenced study outcomes. The studies in this review had durations varying from three days to one year but were, in general, relatively short (3–6 months). Therefore, we cannot exclude that longer-term nutritional interventions could result in more beneficial effects for asthma patients. Furthermore, pathophysiologic mechanisms differ between asthma endotypes [90], suggesting that characteristics of the study population (e.g., age, obesity) influence the success of nutritional interventions. We here show the data for the studies presented in this review did not suggest that the success of nutritional interventions was depended on age. However, other characteristics of the study population could have been of influence. Other examples are indications that vitamin C may be relevant in the prevention of viral-induced exacerbations [91], whereas weight loss interventions may be most relevant for the obesity-induced asthma phenotype [86]. Information on asthma phenotypes or endotypes was missing in many of the trials included in this review. A recommendation for future research is to provide this information since it is crucial in order to interpret the effect of nutritional interventions in asthma patients.

In summary, this review provides an overview of studies that examined nutritional interventions in asthma patients and reported changes in asthma-related outcomes as well as immunological parameters. Certain components from the herbs, herbal mixtures, and extract cluster, as well as the omega-3 LCPUFAs, are promising interventions in the context of improving asthma-related outcomes via immunomodulation. Only three interventions showed clinically relevant improvements. Future studies should now focus on how to optimize the beneficial effects of nutritional interventions in asthma patients, e.g., by considering the phenotypes and endotypes of asthma in the study population. The potential of these interventions and underlying pathways should be explored further before any of these interventions could be added to lifestyle guidelines for asthma patients.

## Figures and Tables

**Figure 1 nutrients-12-03839-f001:**
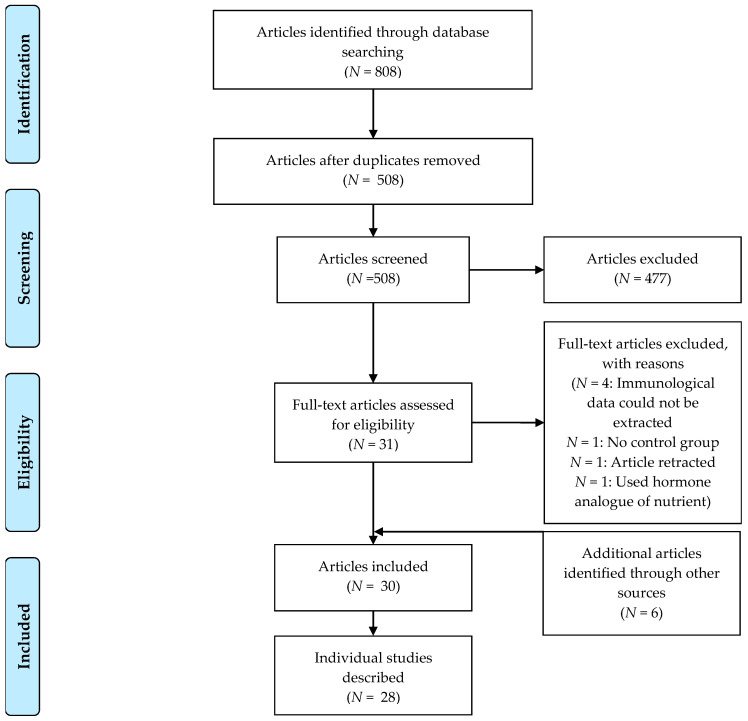
PRISMA flow diagram of study selection. In total, 30 articles describing 28 studies were included in the review.

**Figure 2 nutrients-12-03839-f002:**
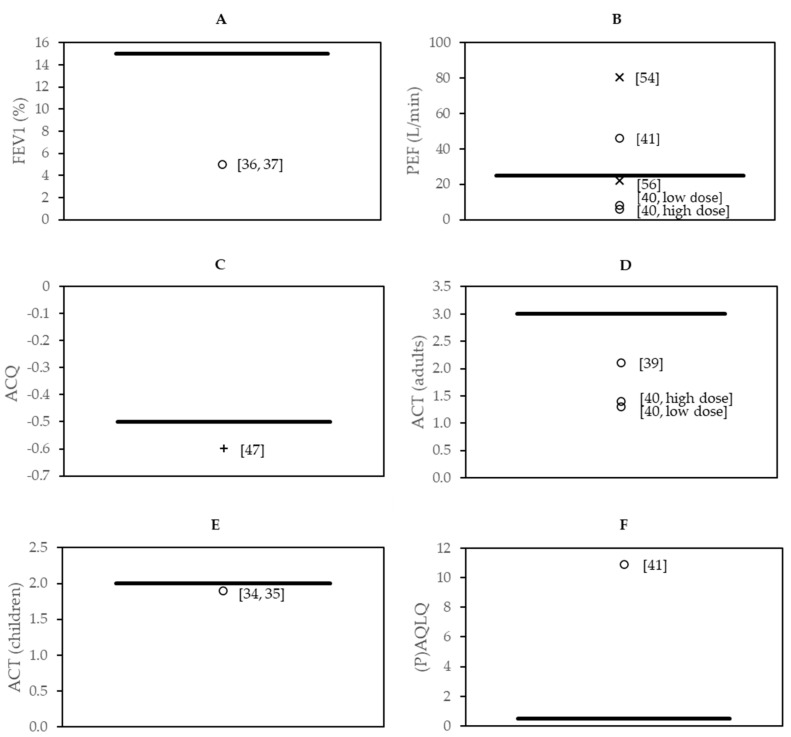
The magnitude of changes in asthma-related outcomes compared to the minimal clinically important difference for (**A**) FEV1; (**B**) PEF; (**C**) ACQ; (**D**) ACT for adults; (E) ACT for children; and (**F**) (P)AQLQ. The symbols (◦ = herbs, herbal mixtures and extracts; x = omega-3 LCPUFA; + = weight loss) represent individual studies, and the line (−) indicates the minimal clinically important difference. Reference numbers for individual studies are noted next to the corresponding symbol.

**Table 1 nutrients-12-03839-t001:** Study characteristics.

Cluster	First Author (Year)	Study Design	Population	Asthma Diagnosis	Intervention and Dose ^##^	Study Duration	*n*	Age (Years)	Male (%)
Herbs, herbal mixtures and extracts	Barlianto (2017) * Barlianto (2018) * [34,35]	Parallel	Children with asthma	GINA guidelines	*Nigella sativa* oil 15–30 mg/kg/day	8 weeks	28	9 ^#^	39 ^#^
	Hosseini (2018) ** Zilaee (2019) ** [36,37]	Parallel	Adults with mild-to-moderate asthma	GINA guidelines	Saffron 100 mg/day	8 weeks	76	41 ^#^	63 ^#^
	Khayyal (2003) [38]	Parallel	Adults with mild-to-moderate asthma	National Institutes of Health and GINA guidelines	Aqueous extract of propolis 13% solution, equivalent to active constituents in 2 mL of aqueous extract of propolis per day	2 months	46	Range: 19–52	78 ^#^
	Koshak (2017) [39]	Parallel	Adults with asthma	GINA guidelines and ACT score	*Nigella sativa* oil 1 g/day	4 weeks	80	41 ^#^	41 ^#^
	Salem (2017) [40]	Parallel, 3 arms	Adults with asthma	Previous physician’s diagnosis and National Institutes of Health criteria	*Nigella sativa* (low dose) 1 g/day	12 weeks	76	38 ^#^	34 ^#^
					*Nigella sativa* (high dose) 2 g/day				
	Yugandhar (2017) [41]	Parallel	Adults with bronchial asthma	Previous physician’s diagnosis	Extract of *B. serrata gum* resin and *A. marmelos* fruit 200 mg/day	56 days	29	39 ^#^	41 ^#^
Supplements	Ghaffari (2014) [42]	Parallel	Children with moderate asthma	Previous physician’s diagnosis	Vitamin E 50 mg/day	8 weeks	240	9 ^#^	54 ^#^
	Pearson (2004) [43]	Parallel	Adults with asthma	Previous physician’s diagnosis and medication use	Vitamin E 500 mg/day	6 weeks	72	48	46
	Smith (2015) [44]	Parallel	Children and adults with asthma	Previous physician’s diagnosis, symptoms and medication use	Soy isoflavone 100 mg/day	6 months	386	36	34
	Wood (2008) [45]	Crossover, 3 arms	Adults with stable asthma	Previous physician’s diagnosis, symptoms and airway hyper-responsiveness	Tomato extract 45 mg lycopene/day	7 days	22	52	36
					Tomato juice 45 mg lycopene/day				
Weight loss	Dias-Junior (2014) [46]	Parallel	Obese adults with severe asthma	Previous physician’s diagnosis and treatment according to GINA guidelines	Low-calorie intake, use of sibutramine (10 mg/day) and use of orlistat (max. 120 mg/day)	6 months	33	43 ^#^	6 ^#^
	Jensen (2013) [47]	Parallel	Obese children with asthma	Previous physician’s diagnosis	Energy reduction (−500 kcal/day) and counseling sessions	10 weeks	28	12 ^#^	61 ^#^
	Toennesen (2018) ^1^ [48]	Parallel, 4 arms	Adults with asthma	ACQ score and positive diagnostic test	High protein and low glycemic index diet	8 weeks	125	40 ^#^	31 ^#^
					Combination of diet and exercise				
Vitamin D3	Bar Yoseph (2015) [49]	Parallel	Children with mild asthma	Previous physician’s diagnosis, positive methacholine challenge test	Vitamin D3 14.000 IU/week	6 weeks	39	13 ^#^	64 ^#^
	Castro (2014) [50]	Parallel	Adults with symptomatic asthma	Previous physician’s diagnosis, evidence of bronchodilator reversibility or airway hyper-responsiveness	Vitamin D3 100.000 IU once, followed by 4000 IU/day	28 weeks	408	40 ^#^	32 ^#^
	de Groot (2015) [51]	Parallel	Adults with nonatopic asthma	Evidence of bronchodilator reversibility or airway hyper-responsiveness	Vitamin D3 (cholecalciferol) 400.000 IU single dose	9 weeks	44	56 ^#^	59 ^#^
	Kerley (2016) [52]	Parallel	Children with uncontrolled asthma	Previous physician’s diagnosis and medication use according to GINA guidelines	Vitamin D3 2000 IU/day	15 weeks	39	8 ^#^	62 ^#^
	Martineau (2015) [53]	Parallel	Adults with asthma	Previous physician’s diagnosis, evidence of bronchodilator reversibility	Vitamin D3 (Vigantol oil) 120.000 IU/2 months	1 year	250	48 ^#^	44 ^#^
Omega-3 LCPUFA	Emelyanov (2002) [54]	Parallel	Adults with mild-to-moderate atopic asthma	American Thoracic Society asthma definition	Lipid extract of the New Zealand green-lipped mussel 200 mg/day EPA + DHA	8 weeks	46	39 ^#^	26 ^#^
	Hodge (1998) [55]	Parallel	Children with asthma and a history of episodic wheeze in the last 12 months and airway hyperresponsiveness to histamine	Symptoms and airway hyper-responsiveness	Omega-3 fatty acid-rich diet and omega-3 fatty acid supplementation 1200 mg/day EPA + DHA	6 months	39	10 ^#^	41 ^#^
	Mickleborough (2013) [56]	Crossover	Adults with mild-to-moderate persistent asthma	Previous physician’s diagnosis	Marine lipid fraction PCSO-524™ 400 mg/day omega-3 LCPUFA, of which 120 mg/day EPA + DHA	3 weeks	20	23	60
	Moreira (2007) [57]	Parallel	Adults with stable, persistent asthma	Previous physician’s diagnosis and use of inhaled corticosteroids	N-3 PUFA 780 mg/day EPA+DHA 10 mg/day vitamin E	2 weeks	20	38 ^#^	0
	Schubert (2009) ^2^ [58]	Parallel	Adults with asthma and house dust mite allergy	Unknown	N-3 PUFA-enriched fat blend 750 mg/day (of which 630 mg/day EPA + DHA)	3 weeks	23	24 ^#^	43 ^#^
Whole food approaches	Bseikri (2018) [59]	Parallel	Obese adolescents with asthma	Previous physician’s diagnosis and ACQ score	Nutrient-dense bar (CHORI-bar) 2 bars/day	2 months	56	15 ^#^	55 ^#^
	Papamichael (2019) [60]	Parallel	Children with mild asthma	Previous physician’s diagnosis and GINA guidelines	Two meals with fatty fish per week as part of the Greek Mediterranean diet	6 months	64	8 ^#^	52 ^#^
	Sexton (2013) [61]	Parallel, 3 arms	Adults with symptomatic asthma	Previous physician’s diagnosis, bronchodilator reversibility or PEFR variability during run-in	High-intervention: encouraged to adopt a Mediterranean diet and received intensive initial advice and 41 h of consultation sessions with a dietitian	12 weeks	35	38 ^#^	29 ^#^
					Low intervention: received less intensive advice and spent 2 h with a dietitian				
	Sudini (2016) [62]	Parallel	Adults with asthma and a positive skin test to an indoor allergen	Previous physician’s diagnosis	Broccoli sprouts 100 g/day	3 days	40	34 ^#^	40 ^#^
	Wood (2012) ^2^ [63]	Parallel	Adults with stable asthma	Previous physician’s diagnosis, symptoms and airway hyper-responsiveness	High antioxidant diet	14 days	137	57 ^#^	42 ^#^

Abbreviations: ACT = asthma control test; ACQ = asthma control questionnaire; DHA = docosahexaenoic acid; EPA = eicosapentaenoic acid; GINA = Global Initiative for Asthma; IU = international unit; (LC)PUFA = long-chain polyunsaturated fatty acid; PEFR = peak expiratory flow rate; ^1^ = exercise arm not included in this review; ^2^ = not all data of this article could be extracted; */** = articles were based on the same study; ^#^ = calculated; ^##^ = habitual intake and reference data for nutritional interventions is presented in Appendix A.

**Table 2 nutrients-12-03839-t002:** Changes in asthma-related outcomes and immunological parameters as compared to the control group for the herbs, herbal mixtures and extracts cluster.

First Author (Year)	Jadad Score ***	Intervention and Dose	Asthma-Related Outcomes					Immunological Parameters					
			Lung Function			Asthma Control	QoL						
			FEV_1_	FVC	PEF	ACT	(P)AQLQ	FeNO	Cells (Sputum, Blood)	Th1	Th2, IgE	Treg	Pro-Inflammatory Markers
Barlianto (2017) * Barlianto (2018) * [34,35]	2	*Nigella sativa* oil 15–30 mg/kg/day				↑ ^1^				↑	↓		
Hosseini (2018) ** Zilaee (2019) ** [36,37]	5	Saffron 100 mg/day	↑	↑					=				↓
Khayyal (2003) [38]	2	Aqueous extract of propolis 13% solution, equivalent to active constituents in 2 mL of aqueous extract of propolis per day	N/A	N/A	N/A					N/A		N/A	N/A
Koshak (2017) [39]	5	*Nigella sativa* oil 1 g/day	=		=	↑			↓		=		
Salem (2017) [40]	3	*Nigella sativa* (low dose) 1 g/day	=	=	↑	↑		=		↑	=	=	=
		*Nigella sativa* (high dose) 2 g/day	=	=	↑	↑		=		↑	=	=	=
Yugandhar (2017) [41]	4	Extract of *B. serrata gum* resin and *A. marmelos* fruit 200 mg/day	=		↑		↑			↑	↓		

Abbreviations: FEV_1_ = forced expiratory flow in one second; FVC = forced vital capacity; PEF = peak expiratory flow; ACT = asthma control test; (P)AQLQ = (pediatric) asthma quality of life questionnaire; FeNO = fractional exhaled nitric oxide; Th = T helper cell; IgE = immunoglobulin E; Treg = regulatory T cell; QoL = quality of life; ↑ = significant increase as compared to the control group; ↓ = significant decrease as compared to the control group; = = no change as compared to the control group; N/A = between-group changes not reported; * = based on the same study; ** = based on the same study; *** = Jadad score was calculated to assess the methodological quality of the RCTs. The calculation of the Jadad score can be found in Appendix A; ^1^ = both articles presented ACT data; only between-group changes are shown here since comparing changes results in less bias than comparing scores at the final time point. Quantitative data showing the within-group changes is presented in Appendix A.

**Table 3 nutrients-12-03839-t003:** Changes in asthma-related outcomes and immunological parameters as compared to the control group for the supplements cluster.

First Author (Year)	Jadad Score *	Intervention and Dose	Asthma-Related Outcomes					Immunological Parameters			
			Lung Function			Asthma Control					
			FEV_1_	FVC	PEF	ACT	ACQ	FeNO	Cells (Sputum, Blood)	Th2, IgE	Pro-Inflammatory Markers
Ghaffari (2014) [42]	3	Vitamin E 50 mg/day	N/A	N/A						N/A	
Pearson (2004) [43]	5	Vitamin E 500 mg/day	=	=	=					=	
Smith (2015) [44]	5	Soy isoflavone 100 mg/day	=	↓	=	=		↑	=		=
Wood (2008) [45]	3	Tomato extract 45 mg lycopene/day	=	=			=	=	↓; =		
		Tomato juice 45 mg lycopene/day	=	=			=	=	↓; =		

Abbreviations: FEV_1_ = forced expiratory flow in one second; FVC = forced vital capacity; PEF = peak expiratory flow; ACT = asthma control test; ACQ = asthma control questionnaire; FeNO = fractional exhaled nitric oxide; Th = T helper cell; IgE = immunoglobulin E; ↑ = significant increase as compared to the control group; ↓ = significant decrease as compared to the control group; = = no change as compared to the control group; N/A = between-group changes not reported; * = Jadad score was calculated to assess the methodological quality of the RCTs. The calculation of the Jadad score can be found in Appendix A. Quantitative data showing the within-group changes is presented in Appendix A.

**Table 4 nutrients-12-03839-t004:** Changes in asthma-related outcomes and immunological parameters as compared to the control group for the weight loss cluster.

First Author (Year)	Jadad Score *	Intervention and Dose	Asthma-Related Outcomes					Immunological Parameters			
			Lung Function		Asthma Control		QoL				
			FEV_1_	FVC	ACT	ACQ	(P)AQLQ	FeNO	Cells (Sputum, Blood)	Th2, IgE	Pro-Inflammatory Markers
Dias-Junior (2014) [46]	3	Low-calorie intake, use of sibutramine (10 mg/day) and use of orlistat (max. 120 mg/day)	=	↑	↑	↓		=	=	=	=
Jensen (2013) [47]	3	Energy reduction (−500 kcal/day) and counseling sessions	=	=		↓	=	=	↓; =		↓; =
Toennesen (2018) [48]	3	High protein and low glycemic index diet	=	=		=	=	=	=		=
		Combination of diet and exercise	=	=		↓	↑	=	=		=

Abbreviations: FEV_1_ = forced expiratory flow in one second; FVC = forced vital capacity; ACT = asthma control test; ACQ = asthma control questionnaire; (P)AQLQ = (pediatric) asthma quality of life questionnaire; FeNO = fractional exhaled nitric oxide; Th = T helper cell; IgE = immunoglobulin E; QoL = quality of life; ↑ = significant increase as compared to the control group; ↓ = significant decrease as compared to the control group; = = no change as compared to the control group; * = Jadad score was calculated to assess the methodological quality of the RCTs. The calculation of the Jadad score can be found in Appendix A. Quantitative data showing the within-group changes is presented in Appendix A.

**Table 5 nutrients-12-03839-t005:** Changes in asthma-related outcomes and immunological parameters as compared to the control group for the vitamin D3 cluster.

First Author (Year)	Jadad Score *	Intervention and Dose	Asthma-Related Outcomes							Immunological Parameters				
			Lung Function			Asthma Control		QoL						
			FEV_1_	FVC	PEF	ACT	ACQ	(P)AQLQ	Other	FeNO	Cells (Sputum, Blood)	Th2, IgE	Treg	Pro-Inflammatory Markers
Bar Yoseph (2015) [49]	4	Vitamin D3 14.000 IU/week							= ^1^	=	=	=		=
Castro (2014) [50]	5	Vitamin D3 100.000 IU once, followed by 4000 IU/day	=			=					=			
de Groot (2015) [51]	4	Vitamin D3 (Cholecalciferol) 400.000 IU single dose	=				=	=		=	↓ ^2^	=		
Kerley (2016) [52]	3	Vitamin D3 2000 IU/day	=	=		=		=					=	↑
Martineau (2015) [53]	5	Vitamin D3 (Vigantol oil) 120.000 IU/2 months	=		=	=				=				

Abbreviations: FEV_1_ = forced expiratory flow in one second; FVC = forced vital capacity; PEF = peak expiratory flow; ACT = asthma control test; ACQ = asthma control questionnaire; (P)AQLQ = (pediatric) asthma quality of life questionnaire; FeNO = fractional exhaled nitric oxide; Th = T helper cell; IgE = immunoglobulin E; Treg = regulatory T cell; QoL = quality of life; ↑ = significant increase as compared to the control group; ↓ = significant decrease as compared to the control group; = = no change as compared to the control group; * = Jadad score was calculated to assess the methodological quality of the RCTs. The calculation of the Jadad score can be found in Appendix A; ^1^ = provocative dose causing a 20% fall in FEV_1_; ^2^ = significant difference after correction for baseline. Quantitative data showing the within-group changes is presented in Appendix A.

**Table 6 nutrients-12-03839-t006:** Changes in asthma-related outcomes and immunological parameters as compared to the control group for the omega-3 long-chain polyunsaturated fatty acids (LCPUFA) cluster.

First Author (Year)	Jadad Score *	Intervention and Dose	Asthma-Related Outcomes			Immunological Parameters		
			Lung Function		Asthma Control			
			FEV_1_	PEF	ACQ	FeNO	Cells (Sputum, Blood)	Other
Emelyanov (2002) [54]	5	Lipid extract of the New Zealand green-lipped mussel 200 mg/day EPA+DHA	=	↑; =				↓ ^1^
Hodge (1998) [55]	4	Omega-3 fatty acid-rich diet and omega-3 fatty acid supplementation 1200 mg/day EPA+DHA	=				=	
Mickleborough (2013) [56]	5	Marine lipid fraction PCSO-524™ 400 mg/day omega-3 LCPUFA, of which 120 mg/day EPA+DHA		↑		↓		
Moreira (2007) [57]	5	N-3 PUFA 780 mg/day EPA+DHA 10 mg/day vitamin E	=		=	=		
Schubert (2009) [58]	4	N-3 PUFA-enriched fat blend 750 mg/day (of which 630 mg/day EPA+DHA)	=			↓	=	

Abbreviations: FEV_1_ = forced expiratory flow in one second; PEF = peak expiratory flow; ACQ = asthma control questionnaire; FeNO = fractional exhaled nitric oxide; ↑ = significant increase as compared to the control group; ↓ = significant decrease as compared to the control group; = = no change as compared to the control group; * = Jadad score was calculated to assess the methodological quality of the RCTs. The calculation of the Jadad score can be found in Appendix A; ^1^ = exhaled H_2_O_2_. Quantitative data showing the within-group changes is presented in Appendix A.

**Table 7 nutrients-12-03839-t007:** Changes in asthma-related outcomes and immunological parameters as compared to the control group for the whole-food approach cluster.

First Author (Year)	Jadad Score *	Intervention and Dose	Asthma-Related Outcomes						Immunological Parameters					
			Lung Function			Asthma Control		QoL						
			FEV_1_	FVC	PEF	ACT	ACQ	(P)AQLQ	FeNO	Cells (Sputum, Blood)	Th1	Th2, IgE	Treg	Pro-Inflammatory Markers
Bseikri (2018) [59]	2	Nutrient-dense bar (CHORI-bar) 2 bars/day	=	=		=			=			=		=
Papamichael (2019) [60]	3	Mediterranean diet	=	=	=		=	=	↓ ^1^					
Sexton (2013) [61]	2	Mediterranean diet (high intervention) ^2^	=	=			=	=		=	=		=	=
		Mediterranean diet (low intervention) ^2^	=	=			=	=		=	=		=	=
Sudini (2016) [62]	4	Broccoli sprouts 100 g/day	=	=		=			=			=		=
Wood (2012) [63]	3	High antioxidant diet	↑	↑			=		=	=	=			=

Abbreviations: FEV_1_ = forced expiratory flow in one second; FVC = forced vital capacity; PEF = peak expiratory flow; ACT = asthma control test; ACQ = asthma control questionnaire; (P)AQLQ = (pediatric) asthma quality of life questionnaire; FeNO = fractional exhaled nitric oxide; Th = T helper cell; IgE = immunoglobulin E; Treg = regulatory T cell; QoL = quality of life; ↑ = significant increase as compared to the control group; ↓ = significant decrease as compared to the control group; = = no change as compared to the control group; * = Jadad score was calculated to assess the methodological quality of the RCTs. The calculation of the Jadad score can be found in Appendix A; ^1^ = significant difference after correction for baseline; ^2^ = High-intervention: encouraged to adopt a Mediterranean diet and received intensive initial advice and 41 h of consultation sessions with a dietitian. Low intervention: received less intensive advice and spent 2 h with a dietitian. Quantitative data showing the within-group changes is presented in Appendix A.

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
