# Peer review of "Nutritional Interventions to Improve Asthma-Related Outcomes through Immunomodulation: A Systematic Review"

_nutrients, 2020, doi:10.3390/nu12123839_

Round 1
Reviewer 1 Report
Authors wished to investigate through a systematic review whether nutritional interventions may improve asthma-related outcomes and whether these changes occur via immunomodulation. They identified 28 studies which met the inclusion criteria, six exploring herbs, herbal mixtures and extracts, four supplements, three weight loss, five vitamin D3 , five omega-3 long-chain polyunsaturated fatty acids (LCPUFAs) and five whole food approaches.
Herbal mixture as well as the omega-3 LCPUFAs appeared the most promising interventions in the context of improving asthma related outcomes via immunomodulation (decrease Th2 response).
The review is well-written and useful for clinicians.
Minor point
I suggest Authors to underline the short duration of the studies (generally less than 3-6 months). It is possible that a longer duration of dietary intervention is needed before observing useful results.
Author Response
Question: The review is well-written and useful for clinicians. I suggest Authors to underline the short duration of the studies (generally less than 3-6 months). It is possible that a longer duration of dietary intervention is needed before observing useful results.
Answer: Thank you for your kind words and for this suggestion. We have now added a statement with regard to study duration to the discussion (page 28, line 483-486).
Reviewer 2 Report
Compliments to the authors for this comprehensive and complex topic in a disease as complex as asthma. I would just like to ask if it was evaluated and if threre was any difference for any nutritional interventions based on the age of the patients. For example, if the younger subjects had some advantage over the older ones.
Thanks and best regards
Author Response
Question: Compliments to the authors for this comprehensive and complex topic in a disease as complex as asthma. I would just like to ask if it was evaluated and if there was any difference for any nutritional interventions based on the age of the patients. For example, if the younger subjects had some advantage over the older ones.
Answer: Thank you for the compliments and the question. We indeed evaluated if there was a difference in changes in asthma-related outcomes and immunological parameters based on the age of the study population. However, we did not detect a difference. We have now added this to the discussion (page 28, line 488-491).
This manuscript is a resubmission of an earlier submission. The following is a list of the peer review reports and author responses from that submission.
Round 1
Reviewer 1 Report
Summary: Brakel and colleagues evaluated various studies and clinical trials with nutritional interventions that reported both asthma-related outcomes as well as immunological parameters with an overarching goal to better understand these mechanisms. The authors suggest that beneficial effects on asthma-related outcomes by certain nutritional interventions are mediated by the immune system. I appreciate the authors for their effort in summarizing the comprehensive research systematically. Few specific comments are presented below for improvement:
Major comments:
- Abstract: Line 18-20: The total studies analyzed sums to 33 (8+6+3+6+6+4) and not 32 studies as mentioned. Please check.
- Abstract, Line 21-25: findings have been discussed sequentially but stratification/presentation of findings could be improved. One suggestion could be to put studies showing improvements together, and then say, of those nine studies reported improvements in both asthma-related outcomes and immunological parameters followed by studies reporting worsening of asthma-related outcomes or immunological parameters.
- The authors state that systematic evaluation of randomized controlled trials studying various nutritional interventions could improve asthma related options. However, it isn’t clear what the findings may mean on a broader scale because there is no one recipe for asthma considering different asthma phenotypes and endotypes. Is the author trying to suggest in the discussion that different nutritional interventions could be useful for different asthma phenotypes? Could the authors elaborate own thoughts and provide perspective on how this evaluation could be useful or what improvements would author suggest/speculate as preventative measures for better asthma control?
- Did the authors include or at least check other cross-sectional and longitudinal studies apart from clinical trials? This is to ensure that by focusing on RCTs, the authors have not overlooked some other possible interesting analyses.
Minor comments
1. Line 54-56, Add reference after Global Initiative for Asthma (GINA).
2. Line 57-59: Reference 17 is for GINA guidelines. Are there any studies (not reviews), which have identified specifically that quitting smoking, loosing weight, physical activity, diet of fruits and vegetables etc. improved asthma outcomes? It would be nice to cite them.
3. Line 117-118 in Methods section: I think the sentence ending with “different studies” is incomplete.
4. Line 142, Results section: “with respective the number of articles”. It should be either “with their respective number of articles” or “with respect to number of articles”. May be rephrase/clarify?
5. Line 143: Specify vitamin D2 or vitamin D3? Make this consistent
Reviewer 2 Report
General comments:
I was expecting to learn something by reading this manuscript. But alas, all I ended up with was the same general sense of things I could have pulled together relatively quickly by searching Pubmed.
The authors obviously went to a lot of effort to pull all this material together, but in the end, this manuscript presents itself more as an introductory chapter to a PhD thesis, than as an overview of a topic.
What is missing is any effort at assimilation of quantitative data in this review. Table 2, which is the main part of the results, shows only qualitative outcomes, and with no effort to weight the interpretation for sample size.
Reviewer 3 Report
This is an interesting study exploring links between nutritional interventions, asthma and the immune response.
My major reservations are as follows
The authors do not provide concrete, discrete estimations of effect size - they say, for example, that an intervention "improved" an asthma parameter - but not by how much, or the range. They do provide a table - table 3- that I believe is meant to summarise this findings- but I find this is not a clear presentation at all. The results and discussion need to provide much clearer evidence on the range of the stated improvement, and clinical relevance as well as statistical significance.
My other concern is that no attempt has been made to grade the evidence - the number of study subjects and duration are provided but no analysis of drop out rates, blinding etc, evidence of publication bias etc have been provided
In addition, we are not given a date range for selected studies, nor (that I saw) whether they used only English language studies, and whether study publication in a peer reviewed journal was one of the inclusion criteria. Also, although FEV1 is used to assess asthma severity and treatment response, did the authors check the FEV1:FVC ratio of published works as it is that ratio that actually defines that a person has an obstructive airways disease - in other words how did studies actually ensure that enrolled patients had asthma?
The authors have also combined studies in children and in adults - can they provide evidence that asthma behaves the same way in these two different populations
Lastly, the discussion in parts feels as if it repeats the results - the findings of studies with conflicting results are not explained in depth, and the significance of some studied parameters not discussed at all - for example what is the immunological significance of FeNO concentrations and can references be provided that support that changes in this parameter reflect immune modulation in the context of asthma.
Other specific points follow:
Abstract should specify studies evaluated in the methods
Introduction - several statements are not supported by references - see line 33, 34
line 36 - for what country are these recommendations? Also, the reference for non-adherence to inhaled steroids is a qualitative not quantitative reference - can the authors provide quantitative evidence? Also, asthma recoemmdations for children are different compared with adults - in, for example, when to add leukotriene antagonists - can the authors provide references for specific treatment protocols.
Also, generally throughout the intro please specify study type
lines 106-109 - can the authors provide a reference that this is an acceptable way to manage this part of the analysis
line 142- check grammar
in the table with study characteristics - what is in the herbal formulations?
what is meant by "high" vs "low" intervention in the Sexton study?
Regarding results - please see my comments under major concerns
In the discussion there needs to be an expansion of the clinical relevance of findings - if one parameter is improved which is it, and how much does it matter if it is the only parameter improved?
I don't think the conclusion - as the manuscript is presented/written- supports the findings.
Round 2
Reviewer 2 Report
General comments
It looks like I was the most negative of the three reviewers of the previous version of this manuscript. I did not spend much time on it then, because I felt it was a relatively arbitrary cataloguing of things I could have looked up myself on PubMed or on Google Scholar. While this type of manuscript is not to my liking, I can see that some people might benefit from this sort of summary.
Overall: There are far too many unnecessary abbreviations used in this manuscript, and that makes this mindbogglingly difficult item even more difficult to to read. For example look at line hundred 348 paragraph 3.7. Could any reader of that paragraph come away with any more information than to simply be bewildered? Be aware that high-level journals discourage abbreviations..
Specific comments
1. Has any reader of this or the earlier version of this manuscript even looked at the references? Most of them do not specify which journal each citation comes from. (As a suggestion I highly recommend Zotero as a very convenient and easy to use method for doing references.)
2. line 352 mentions Figure 2, whereas it appears Table 3 would be more appropriate. Importantly, the contents of Table 3 are indecipherable and the values mean nothing to me. As a suggestion specify what the minimal clinical important difference is for each item and then indicate by how much the intervention affected that.
3. Table 1. If one bears in mind that many drugs are versions of herbal remedies (eg ASA), it should be appreciated that not every herb should be considered nutrition. Dosage is also important and if the clinical trials reviewed in this table and in the manuscript use intakes beyond what might be contained in a typical diet, then the dosage must be specified for every one of the clinical trials reviewed.
This critique is particularly relevant to vitamin D because the introduction to this manuscript ( line 79 ) states that a shortcoming of other similar reviews is that dosage, (and likewise treatment duration) had not been considered. This makes the reader expect that the authors will address the shortcoming they point out for other papers on this topic they themselves are writing about.
3b The clinical trial of Ali 2017 (and at Line 281) must be eliminated from this manuscript because alpha-calcidiol is a toxic and potent drug formulation and not relevant as a nutrient form of vitamin D. (for example, is cortisol a cholesterol analogue, and hence a nutrient?, if not, then likewise for alpha-calcidiol)
4. Line 193. It is picky, but the authors should be aware of this kind of error: "one study" is not the subject of a sentence whose verb is "evaluated".
5. Line 198. The Chinese herbal remedies presented are ambiguous compounds and hardly suitable for readers of nutrition research. These aspects would be more suitable for a journal of complementary medicine.
6. In terms of weight loss intervention, Okoniewski et al. (AnnalsATS Volume 16 Number 5 May 2019) should be cited, since readers will find that to be a more rigorous examination of the topic presented here.
7. Table 2 is an onerous multipage table that becomes impossible to understand after the first page. This table should be divided up into multiple tables: i.e. one table for the herbs another table for the supplements, another for weight loss, another for vitamin D etc.. Also, include dosage here, and for reference, along with that, provide reference data for the amount that might be the range of the compound found in a normal diet.
8. This is essentially a narrative review that comes to essentially no conclusion. If the work is to be useful beyond a simply cataloguing, then there should be more discussion as to what the gaps are in the knowledge and where further research is required.
It should also be worthwhile to consider whether the evidence to this point is so strong that it is unlikely for any of these nutrients to have any bearing on asthma. i.e. is there any certainty that are any one of these nutrients is not useful?
9. The references, starting at line 555. Many of these do not have the journal name included. Reference 16 doesn't even have much there.
10. More comment than critique.... Line 152, and Ref 133. The Jadad score is a trivial way of assessing “quality” of something. It was designed so that a non-expert can give a sense of quality on a clinical trial manuscript that matches the scoring of an expert. I do not see how the Ali paper could get only a score of only 1 out of 5. But then, the scoring system is arbitrary, eg was the word "blind" found in the publication? If not, it gets a lower score.
